# A Study of the Mechanical Properties in Composite Materials with a Dammar Based Hybrid Matrix and Reinforcement from Crushed Shells of Sunflower Seeds

**DOI:** 10.3390/polym14030392

**Published:** 2022-01-19

**Authors:** Marius Marinel Stănescu, Alexandru Bolcu

**Affiliations:** Department of Mechanics, University of Craiova, 165 Calea Bucureşti, 200620 Craiova, Romania; mamas1967@gmail.com

**Keywords:** hybrid resin, sunflower seed shells, composite materials, mechanical properties

## Abstract

The production of composite materials through the reuse of waste or by-products from the agri-food industry will be a challenge for environmental protection. This study focuses in that direction. In the first stage, composites were made with the hybrid resin matrix (with three major volume proportions of natural Dammar resin and epoxy resin) and the reinforcing from shredded shells of sunflower seeds. Based on the tensile and compressive stresses, the stress–strain and strain–strain diagrams were obtained. The surface area of the rupture was investigated with stereo-microscopic analysis, and the absorption/loss of water was studied with a high precision balance. The vibration behavior was investigated experimentally, determining the damping coefficient and its own frequency. In the second stage, the study of these materials was extended. Sandwich composites were made with the same type of hybrid matrix as in the first stage. The core was made of shredded shells of sunflower seeds and the outer faces of linen fabric. These composites were applied to the bend (in three points), being obtained the force-deformation diagrams. The determined mechanical properties allow the complete or partial realization of these composites of some furniture components or of some equipment used in the field of constructions.

## 1. Introduction

Society’s development has led to the widespread use of plastic due to the possibility of obtaining cheap finished products. This has also led to a continuous increase in the amount of waste that is difficult to recycle and has harmful effects on the environment [1]. Therefore, reducing the use of petroleum derivatives is one of the current concerns of polymer science [2,3]. The production of composite materials based on natural resins and fibers reduces the use of non-biodegradable polymeric materials and creates environmentally friendly alternatives. In order to increase the ecological value of the final product, the production of fibers obtained from organic farming, or from residual materials such as wheat or rice straw, flax, cane [4,5,6] or from oil palm empty fruit bunches [7], has increased. Obtaining high-performance composites involves the use of long natural fibers that usually require specific procedures that lead to additional energy consumption [8].

Ecological concern and new environmental protection regulations have required the replacement of inorganic fillers with natural fillers in polymer composites. The reuse of waste or by-products in agro-food industry can increase economic value and environmental benefits and better highlight sustainability [9,10,11,12]. The main reasons for the use of natural fillers in the development of polymer composites were: low cost, sustainability [13], low density, non-toxicity and the fact that they are less abrasive during processing [14]. The incorporation of organic particles into a heat-resistant or thermoplastic composite matrix allows a significant reduction in the price of the final product, while obtaining a change in their mechanical behavior [15,16], as well as in their physical [17] and thermal [18] properties. Several types of agricultural waste fillers in the shape of particles were investigated, such as: stems and straw (sunflower) [18], wheat [19], rapeseed [20]); shells (rice) [21,22], sunflower [23]), and pineapple leaves/kenaf fiber [24]. The choice depends on the local resources and the need to eliminate surplus biomass. The hardening efficiency of natural fillers results not only from their geometry, but also from their chemical composition and their cellulose, lignin, starch, protein and lipid content, which influences the disintegration capacity, hardness and rigidity, as well as the adhesion to selected polymers [25].

A very valuable agricultural product is sunflower seed (*Helianthus annulus* L.) from which edible oil and other nutritious foods are obtained [26]. However, after the industrial processing of sunflower seeds, their shells are considered as agricultural residues. Because about 30% of the weight of a sunflower seed is the shell [27], millions of tons of sunflower seed shell residue result annually. The use of this low-value agricultural waste for the development of composite materials offers economic benefits and also contributes to the protection of the environment [26]. Powder from sunflower seed hulls can give the appearance of wood to polymer composites and can thus contribute to the conservation of forest resources.

Studies on the moistening, hardness and surface roughness for polypropylene (PP) composites reinforced with sunflower hull powder in proportions of 30%, 40%, 50% and 60% by weight were made by Kaymakci et al. [28]. The mechanical properties, dimensional stability and crystallization behavior for the same types of composites were investigated by Ayrilmis et al. [29]. In addition to these properties, in [18], the bending and traction behavior for polypropylene composite reinforced with sunflower powder in proportions of 30%, 40%, 50% and 60% by weight was studied. The same properties, but also the loss factor and thermal properties of very low-density polyethylene (PE) composites, containing various amounts of sunflower hulls, were determined by Barczewski et al. [30]. The properties of epoxy resin composites filled with three types of agricultural waste (sunflower shell, walnut shell and hazelnut shell) were studied by Barczewski et al. [31]. Sunflower shells at three percent by weight (10%, 20% and 30% by weight) were used by Vold et al. [32] to produce polyamide (PA) composites. Tensile, impact, bending and moisture absorption properties for these composites were determined in [32].

Kuram [33] studied the effect of the amount of shell filling from sunflower seeds (5%, 10%, 15% and 20% by weight) on the rheological and mechanical properties of acrylonitrile butadiene-styrene (ABS) terpolymer. It was found that the addition of lingo cellulosic filler to the thermoplastic matrix increased the rigidity of the composites. The increase in the proportion of shell filling from sunflower seeds affected the behavior of the composites on impact. This was explained by the fact that sunflower seed hulls are mostly characterized by irregular shape and this led to a significantly reduced ability to transfer mechanical stresses during loading [31]. In this way, the shell filling from sunflower seeds caused additional stresses and triggered cracking in the polymer matrix. The increase in the amount of natural polymer filler has led to an increase in the number of stress concentration points and easier propagation of cracks along the interface between the polymer matrix and the reinforcing particles [34].

Marhoon [35] studied the effect of sunflower shell particles with different weight fractions and particle grain sizes, used as reinforcement material, on tensile strength, modulus of elasticity and water absorption for polyurethane matrix composites. Decreasing the particle size of sunflower hulls leads to increased tensile strength, modulus of elasticity and water absorption.

The use of natural resins reinforced with sunflower seed hulls creates opportunities for obtaining new bio composite products. Kanehashi and Ishimura showed that natural resins diluted with solvents form varnishes that need to be combined with synthetic resins for hardening [36,37]. Among the natural resins, Dammar is very well suited to be strengthened. In-depth studies on the structure and chemical composition of Dammar are made by [38,39,40]. In contrast, studies on the mechanical behavior of this natural resin are relatively few. In [41], some mechanical characteristics such as tensile strength, percentage elongation and Young’s modulus were studied. Zakaria and Ahmad [42] studied the behavior of a new Dammar-modified silicone binder that reduces the use of synthetic binder and that has improved and more environmentally friendly properties. In [43], it was studied how the addition of Dammar contributed to the improvement of the mechanical properties of a modified silicone and the optimal ratio between silicone and Dammar was determined, which ensures the best properties for impact, hardness, tensile and adhesion stresses.

Dammar-based hybrid resins were used to obtain composites reinforced with natural fibers. In [44], are studied the mechanical properties for composites made of hybrid resins with three proportions of Dammar reinforced with linen, cotton, hemp, rush or wheat straw. The tensile strength, the modulus of elasticity, but also the damping properties of the vibrations are compared. Composite materials made from hybrid resins based on Dammar, reinforced with waste paper, are studied in [45]. In [46], the influence of some non-uniformities on the mechanical behavior of composite materials with Dammar-based matrix and reinforcement from hemp fabric is studied.

The development of natural fiber-reinforced composites in high-performance applications requires additional information about their dynamic properties, such as damping. Sufficient damping is required to reduce the vibration of the structures, as well as to avoid fatigue fractures [47]. Initial assessments of the damping properties of traditional fiber-reinforced composites were made in [48,49]. Then, several concepts were used to model the damping of classical composites [50,51]. For natural fiber composites, some researchers have experimentally analyzed the damping performance [52,53,54,55]. In [56], a proper analysis is made on the damping properties of fiber-reinforced polymer composites. A broad analysis is made of several factors that determine damping, such as matrix, fiber types, fiber architecture, fiber surface change, and embedded fillings. Additionally, the role of the interfacial region (where shear energy is stored) in improving damping properties is discussed.

In general, three energy dissipation mechanisms are considered [57]. The first considers that the damping force is proportional to the speed, the second mechanism considers that the damping force is proportional to the frequency of vibration, and the third mechanism considers that the damping rates of its own vibration modes depend proportionally to the square of the frequency. Other aspects and mechanisms of the energy dissipation phenomenon and the influence on vibrations for different composite materials are presented in [58,59,60,61].

The mechanical behavior of composite materials with a hybrid resin matrix based on Dammar and the reinforcement of shells shredded by sunflower seeds are studied in this paper. These composites have limited mechanical properties. This disadvantage can be removed if they are used as a core of sandwich composites, with the outer faces of natural fiber fabrics impregnated with the same type of resin as the core. For all types of composite materials made, some mechanical properties were determined, such as modulus of elasticity, tensile strength, elongation at break, using tensile tests, compression, bending. Vibration damping capacity and water absorption/loss were also studied.

## 2. Materials and Methods

### 2.1. Manufacture of Test Specimens

Dammar natural resin is diluted with turpentine and kept in a liquid state in closed containers. The hardening process of the diluted resin is very long, even if it is applied in thin layers. This disadvantage can be removed by initiating polymerization with a synthetic resin, together with the corresponding hardener. In our case, in order to be able to use this diluted resin as a matrix for the composite materials under study, the polymerization process with Resoltech 1050 type epoxy resin (Resoltech SAS, Rousset, France) and the related Resoltech 1055 type hardener was generated. Technical data of the epoxy resin Resoltech 1050/Resoltech 1055 (further labeled with 0), can be found on the manufacturer’s web-site (see [62]). Two types of hybrid resin with volume proportions of 60% and 80% natural Dammar resin were used. The two types of hybrid resin will be labeled with 1 and 2. The mechanical properties of these hybrid resins have been investigated in [45]. As an observation, it should be noted that volumetric proportions higher than 80% Dammar natural resin were also tested, but it was found that the hardening time of the hybrid resin increases greatly. For this reason, for volume proportions of Dammar greater than 80%, the hybrid resin is no longer “interesting” in terms of applications in the field of composite materials.

Using the epoxy resin (labeled with 0) as matrix and the two types of hybrid resin (labeled with 1 and 2) with reinforcement from shredded shells of sunflower seeds, composite material test specimens were made, corresponding to the tensile, compression and bending stresses. It is necessary to mention that the casting was performed at an ambient temperature of 21–23 ${}^{{}^{\circ}}$C, and to ensure a complete polymerization the test specimens were cut 10 days after casting.

For the tensile stress, using the three types of resin specified above as matrix, three plates of composite materials with shredded reinforcements were made from shredded sunflower seeds (Figure 1a). Out of the cast plates, three sets of 15 test specimens are cut (according to the standard [63]) each labeled with: CS0.1-15 for the composite test specimens with the epoxy resin matrix; CS1.1-15 for composite test specimens with hybrid resin matrix type 1; CS2.1-15 for the composite test specimens with the hybrid resin matrix type 2 (Figure 1b). The test specimen sizes were 250 mm long and 25 mm wide, and the thickness was 6.7 mm for the CS0 test specimens, 6.8 mm for the CS1 test specimens and 6.8 mm for the CS2 test specimens. Out of a total of 15 test specimens/set, 2 test specimens/set to weigh the water absorption are used. For the compressive stress, using the same components, three beams with a square section were cast, with the side of the section measuring 21 mm. From each beam, 15 test specimens are cut (according to the standard [64]) labeled with: B0.1-15 for the composite test specimens with the epoxy resin matrix; B1.1-15 for composite test specimens with a hybrid resin matrix of type 1; B2.1-15 for composite test specimens with a hybrid resin matrix of type 2 (Figure 1c).

The mass proportion of crushed shells of sunflower seeds was: 40–41% for CS0 and B0 test specimens; 40–42% for CS1 and B1 test specimens; 42–44% for CS2 and B2 test specimens. The density of the test specimens was: 1.15–1.16 g/cm${}^{3}$ for CS0-x and B0-x; 1.11–1.12 g/cm${}^{3}$ for CS1-x and B1-x; 1.07–1.08 g/cm${}^{3}$ for CS2-x and B2-x.

Figure 1a shows a samples of shredded sunflower seed hulls used as reinforcement for the studied composites. A test specimen of the three sets of test specimens required for traction and compression, respectively, are given in Figure 1b,c.

Because composite materials with Dammar-based epoxy/hybrid resin matrix and reinforcement of shredded sunflower seed shell have limited mechanical properties, it is difficult to find a practical use for them. The problem can be solved if these composites will be used as the core of sandwich composites, with the outer faces made of natural fiber fabrics, impregnated with the same type of resin as the core. In this respect, three plates of the same composite materials were cast in the first stage for the bending stress as for the tensile stress. The plates had the thicknesses of: 13.1 mm for the epoxy resin plate; 13.0 mm for 60% Dammar hybrid resin plate; 13.1 mm for the 80% Dammar hybrid resin plate.

In the second stage, 4 layers of linen fabric impregnated with the same type of resin as that from the plate were applied on both sides of the plates. The mechanical behavior of Dammar-based hybrid matrix composites reinforced with linen fabric is studied in [65]. After applying the outer layers, the thicknesses of the sandwich plates were: 15.1 mm for the epoxy resin plate; 15.0 mm for 60% Dammar hybrid resin plate; and 15.1 mm for the 80% Dammar hybrid resin plate. A number of 10 test specimens were cut from the plates. Epoxy matrix composite test specimens were labeled with ST0.1-10, type 1 hybrid resin matrix composite test specimens were labeled with ST1.1-10, and those with resin matrix composite hybrid type 2 were labeled with ST2.1-10 (Figure 2). The dimensions of these test specimens were 250 mm long, and 50 mm wide. The bending tensile test (3-point) was performed according to ASTM C393-C393M-06 ([66]).

Figure 2 shows a test specimen of sandwich composite material with a hybrid resin matrix based on Dammar, which has a core of crushed sunflower seeds and flax surfaces. This type of test specimen was tested for bending (in 3 points).

Table 1 shows centrally the main characteristics of the sets of test specimens made.

### 2.2. Equipment Used for Tests

The CS0-x, CS1-x and CS2-x test specimens were subjected to the tensile test, and the ST0-x, ST1-x and ST2-x test specimens were tested for bending (in 3 points). The LLOYD Instruments Lrx PLU mechanical test machine (AMETEK Precision Instruments, Meerbusch, Germany) with a maximum force of 2.5 kN, a maximum cross member stroke of 1400 mm, a 50 mm extensometer [67] were used for these requirements. The machine was equipped with bending bodies designed for 3-point testing.

Test specimens B0-x, B1-x and B2-x were subjected to the compression test. This test was performed with the Walter-Bai LF300 universal static and dynamic test machine (Walter+Bai AG, Löhningen, Switzerland ) with the capacity of 300 kN. The loading speed was 250 N/s, and the frequency of the load cycles up to and including 20 Hz.

Stereo-microscopic analysis of fracture surfaces was performed in cross section on the fracture surface. The study was conducted with the Olympus SZX7 Stereo Microscope (GT Vision Ltd., Suffolk, UK), SZ2-ET with Galilean optical system, which has a resolution of up to 600 lines per millimeter and a zoom ratio of 7:1 [68]. The analysis was performed according to ASTM STP 1203 [69].

Water absorption was studied using the Kern ABJ 220-4NM analytical balance (Want Balance Instrument Co. Ltd., Jiangsu, China) with single-cell technology and 0.0001 g weighing accuracy ([70]). The study was conducted over 9 days. Samples from sets CS0, CS1 and CS2, with a length of 100 mm and a width of 10 mm were used. These were placed in Berzelius glasses, in which 100 mL of drinking water were poured, and then covered with aluminum foil. The weighing process ceased when the difference in weight of the test specimens from one day to the next was below 0.05 g.

Vibration analysis was performed with:-A SPIDER 8 data acquisition system to which the NEXUS 2692-A-0I4 signal conditioner (Hottinger Brüel and Kjaer A/S, Virum, Denmark) has been connected;-CATMAN EASY software (Hottinger Brüel and Kjaer Gmbh, Darmstadt, Germany) for data acquisition and processing;-accelerometer with a sensitivity of 0.04 pC/ms${}^{-2}$.

## 3. Results and Discussions

The specimens from sets CS0, CS1 and CS2 were required for traction; vibration behavior and water absorption/loss were analyzed. The specimens from sets B0, B1 and B2 were required for compression, and the specimens from sets ST0, ST1 and ST2 were required for 3-point bending (the manufacture of these test specimens has been described in Section 2.1).

The stress-strain diagram, tensile strength $R_{m}$ (MPa), percentage elongation at break *A* (%) and modulus of elasticity *E* (N/mm${}^{2}$) are obtained from the tensile test. Figure 3 shows the stress–strain diagrams for representative test specimens from sets CS0-x, CS1-x and CS2-x. These diagrams were obtained based on the tensile test.

The average value and standard deviations for modulus of elasticity *E* (N/mm${}^{2}$), tensile strength $R_{m}$ (MPa) and percentage elongation at break *A* (%) for CS0-x, CS1-x and CS2-x test specimen sets are given in Table 2. The following formulas were used:-For the average value $\bar{x}=\frac{\sum_{i=1}^{n}x_{i}}{n}$;-For the linear average deviation ${\bar{d}}_{x}=\frac{\sum_{i=1}^{n}\left|x_{i}-\bar{x}\right|}{n}$;-For the average square deviation $\sigma_{x}=\sqrt{\frac{\sum_{i=1}^{n}{\left(x_{i}-\bar{x}\right)}^{2}}{n}}$,
where *n* is the number of test specimens tested, $x_{i}$ is the experimentally determined value for the test specimen *i*$\left(i=\bar{1,n}\right)$.

It is important to notice that the composite test specimens with an epoxy resin matrix have the highest tensile strength. The tensile strength of the test specimens CS1 and CS2, respectively, is 3/4 and 2/3, respectively, of the tensile strength of the test specimens CS0.

From the compression test were obtained: force–strain diagram, maximum force $F_{max}$ (N), maximum strain $L_{max}$ (mm) and allowable compressive strength $\sigma_{ac}$ (MPa). The force–strain diagrams for representative test specimens from sets B0, B1 and B2 are given in Figure 4. These were obtained on the basis of the compression test.

The average value and the standard deviations for the maximum load force and the maximum deformation for the B0-x, B1-x and B2-x test specimens sets are given in Table 3.

Changing the proportion of Dammar natural resin in the matrix of composite materials implies a significant change in their mechanical behavior. The value of the modulus of elasticity and tensile strength decreases when the volume of Dammar is increased. In the case of elongation at break, there is a decrease in its values in the case of composite materials with hybrid resin compared to the composite with epoxy resin. However, for composite materials with hybrid resin, an increase in the elongation at break can be seen as the volume proportion of Dammar increases.

If the value of the maximum loading force (respectively, the value of the permissible compressive strength) of the test pieces made of composite material with the epoxy resin matrix is taken as a standard, then the value of this force (respectively, the value of the permissive compressive strength) decreases by about three times for the test pieces cast from a composite material with a 60% Dammar hybrid matrix and approximately six times in the case of test pieces made from a composite material with an 80% Dammar hybrid matrix. If it begins from the value of the maximum deformation for the composite material with the epoxy resin matrix, then the value of this deformation rises with the increase of the volume proportion of Dammar in the matrix of the composite materials.

Figure 5 shows images with the breaking surfaces of some representative test specimens from the three sets: CS0 (Figure 5a); CS1 (Figure 5b); and CS2 (Figure 5c). These images were obtained using stereo-microscopic analysis.

One can see that the flow phenomenon does not occur, the rupture is sudden, a phenomenon characteristic of fragile materials with rupture generated at the interface between the reinforcing elements (crushed shells of sunflower seeds) and the resin matrix. The CS0-x test specimens, made of epoxy resin reinforced with shredded sunflower seed shells, has a smoother and brighter appearance, which means that the rupture took place without a “pulling” of the reinforcing elements. By comparison, the CS1-x and CS2-x test specimens have a rougher and more opaque appearance, which is characterized by a rupture by “pulling” from the hybrid resin the fragments of sunflower seed shells. A possible explanation may be the different mechanical behavior of sunflower seed shells compared to epoxy resin and hybrid resin, respectively. Specifically, they adhere better to the hybrid resin. This can be explained by the fact that the hardening time of hybrid resins is longer, which allows sunflower seed shells to be impregnated with resin.

The water absorption in the test specimens from sets CS0, CS1 and CS2 was determined over 7 days. After a saturation of water absorption was found, the test specimens were kept at a controlled temperature of 28–30 ${}^{{}^{\circ}}$C and weighed daily to measure water loss. After 2 days, the initial weight was reached. Table 4 shows the evolution of the absorption and water loss of the three test specimens, respectively.

One can see that in the first 5 days, there was a constant absorption of water from one day to the next. In the last 2 days, the water absorption was negligible. The final water absorption was: 11.3% for the CS0 test piece; 24.2% for the CS1 test piece; and 20.6% for the CS2 test piece. Therefore, test specimens made of composite materials with a hybrid resin matrix have a double water absorption compared to test specimens made of composite material with an epoxy resin matrix.

Vibration behavior analysis provides information about some mechanical properties of composite materials. Beam vibration studies made of composite materials consider models based on different deformation theories that are adapted to evaluate the static and dynamic characteristics of beams. For the study of the vibrations of the beams from composite materials, “first order shear deformation theory” is used, symbolized FSDT, in which it is considered that a flat and normal section on the average fiber before deformation remains flat during deformation, but it is no longer perpendicular to the medium fiber. The limits of this theory have imposed the introduction of higher order shear deformation theories (HSDT), symbolized by HSDT. Comparisons of the results of these theories, for thick beams, are made by Sayyad [71]. The vibrations of the thin beams can be analyzed using the Euler–Bernoulli model in which a flat and normal section on the medium fiber before deformation remains flat and normal on the medium fiber during deformation. The choice of the theory with which the vibrations of a beam are studied depends on its dimensions. The differences between these theories are presented by Augusta Neto and collaborators [72].

The damping coefficient and the own frequency for the test specimens from the set CS0-x, CS1-x and CS2-x were determined experimentally. The studied test specimens were embedded at one end, and the measuring transducer was mounted at the free end. The free length of the test specimens was changed between 140 mm and 220 mm.

Figure 6 shows the way in which the natural frequency and the damping factor are determined for a representative test specimen from the CS2-x set, for the free length of 220 mm.

The damping factor μ, is calculated using the formula ([45]):$$\mu =\frac{1}{t_{2}-t_{1}}\ln\frac{w_{1}}{w_{2}},$$
where

-$w_{1}$ is the value of the first maximum and $w_{2}$ is the value of the second maximum chosen for the calculation of the depreciation factor;-$t_{1}$ is time for the maximum $w_{1}$ and $t_{2}$ is time for the maximum $w_{2}$, from the experimentally recorded diagram used to calculate the damping factor.

The frequency is calculated with the formula:$$\nu =\frac{n}{t_{2}-t_{1}},$$
where *n* is the number of oscillations between $t_{1}$ and $t_{2}$.

Table 5 presents the results of experimental vibration determinations for the representative test specimens of the three types of composite materials reinforced with shredded sunflower seed shells. The presentation values represent the arithmetic average for three measurements.

The damping factor is an element that characterizes the overall damping capacity of a test specimen. The assessment of the vibration damping capacity for the studied materials can be done by determining the loss factor, which is calculated with the relation $\eta =\frac{\mu }{\pi \nu }$ [45]. For the studied composite materials, the average value of the loss factor is:-η=0.0421 for composite CS0;-η=0.0612 for composite CS1;-η=0.0548 for composite CS2.

Based on the bending test in three points, Figure 7 shows the force–strain diagrams for the representative test specimens from sets ST0, ST1 and ST2.

The average value and the standard deviations for the maximum load force $F_{max}$ (N) and the maximum deformation $L_{max}$ (mm), for the sets of test specimens of type ST0-x, ST1-x and ST2-x, are given in Table 6.

From the diagram, it can observe that there is a linear dependence between the loading force and the measured deformation. This dependency exists throughout the stress. The rupture occurs suddenly with the rupture of the core. As in the case of tensile stress, the best resistance is shown by the composite test specimens with the epoxy resin matrix. The maximum forces for test specimens ST1 and ST2 are 4/5 and 3/4, respectively, of the maximum force for the test specimens ST0. These ratios are higher than the ratios of the tensile strength obtained in the case of tensile stress. The explanation is that part of the stress is taken over by the surfaces of the flax fabric.

## 4. Conclusions

Shredded sunflower seed shells are “waste from the production process” of oil factories and the amount obtained is significant. This waste can be an alternative to “wood sawdust”, which is becoming more expensive and more difficult to find due to environmental protection measures (measures that involve strict rules for cutting down trees). The use of seed shells in combination with natural or hybrid resins is an environmentally friendly alternative which is also economically advantageous.

The mechanical behavior of the studied composites depends on the proportion of natural Dammar resin in the hybrid resin used as a matrix. For composites with epoxy resin and 60% Dammar hybrid resin, there is a decrease in the tensile strength compared to simple resins used as matrix (9.5 MPa for the CS0 test specimens compared to 60 MPa for the Resoltech 1050 epoxy resin, 7.1 MPa for the CS1 test specimens compared to 20.2 MPa for the 60% Dammar hybrid resin [45]). In the case of the composite with 80% Dammar hybrid resin matrix, the tensile strength is comparable to that of the resin used (6.3 MPa for the CS2 test specimens compared to 7.2 MPa for the 80% Dammar hybrid resin [45]). This decrease in tensile strength can be explained by the fact that the shredded seed shells have irregular shapes, with sharp corners, which produce stress concentrations and are dampers for crack propagation. The fact that the elongations at break for the three composite materials are close and have much lower values than the elongation at break for the resins used as matrix shows that the seed shells break first.

In the case of the composite with epoxy resin matrix, there is a decrease in the modulus of elasticity of the composite, compared to the modulus of elasticity of the resin used (3024 MPa for the CS0 test specimens compared to 3300 MPa for the Resoltech 1050 epoxy resin). In contrast, in composite materials with a hybrid resin matrix, the addition of seed shells leads to an increase in the modulus of elasticity (2580 MPa for the CS1 test specimens compared to 1720 MPa for the 60% Dammar hybrid resin and 1814 MPa for the CS2 test specimens compared to 835 MPa for the 80% Dammar hybrid resin [45]).

Water absorption/loss occurred because of the shredded sunflower seed shells since both epoxy resin and hybrid resins are not soluble in water. Absorption occurred over a longer period of time (over 5 days), and loss occurs rapidly (within 2 days).

The change in the damping factor depending on the length of the beam is similar to the variation of the vibration frequency. Therefore, one can believe that, among the damping mechanisms presented, the predominant one is the one in which the damping is considered to be proportional to the vibration frequency. The values of the loss factors of the three studied materials show that the composite materials with hybrid resin matrix have superior damping properties to those with epoxy resin matrix. The highest loss factor was obtained for the 60% Dammar hybrid resin composite.

The analysis of the mechanical behavior of composite materials with hybrid resin matrices, reinforced with shredded sunflower seed shells, shows a decrease in their mechanical properties as the proportion of Dammar increases. These composite materials also have a lower tensile strength and modulus of elasticity compared to the epoxy matrix composite. Instead, they have superior vibration damping properties.

The values of the determined mechanical properties allowed the realization in whole or in part of these composites of some furniture components (table tops, paneling, etc.), of some removable formwork or of some supporting beams, used in the field of constructions.

## Figures and Tables

**Figure 1 polymers-14-00392-f001:**
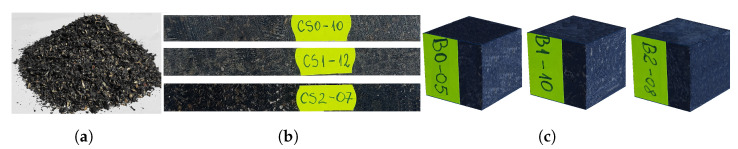
Sample of crushed sunflower seed shells (**a**); test specimens required for traction (**b**); test specimens required for compression (**c**).

**Figure 2 polymers-14-00392-f002:**
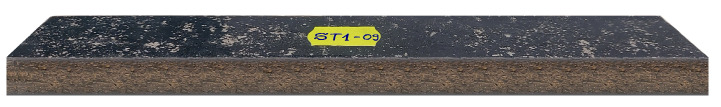
Sandwich composite sandwich test specimen with a 60% Dammar hybrid resin matrix, which has a shredded sunflower seed core and flax fabric surfaces, which was tested for bending in 3 points.

**Figure 3 polymers-14-00392-f003:**
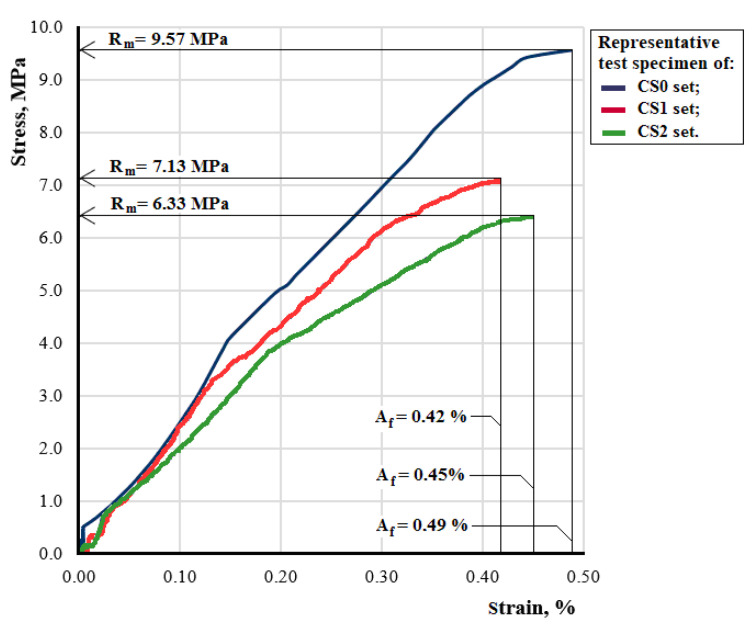
Stress–strain diagram for representative test specimens from sets CS0-x, CS1-x and CS2-x.

**Figure 4 polymers-14-00392-f004:**
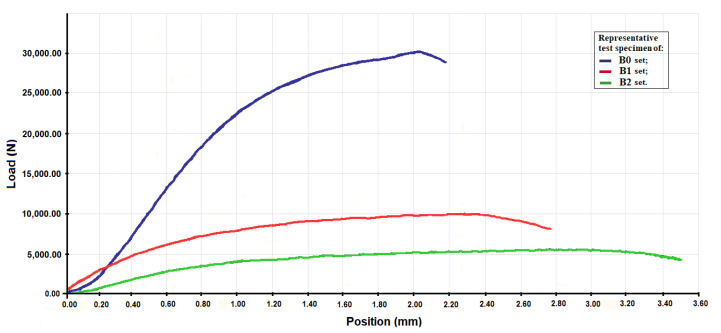
The force–strain diagrams for representative test specimens from sets B0, B1 and B2.

**Figure 5 polymers-14-00392-f005:**
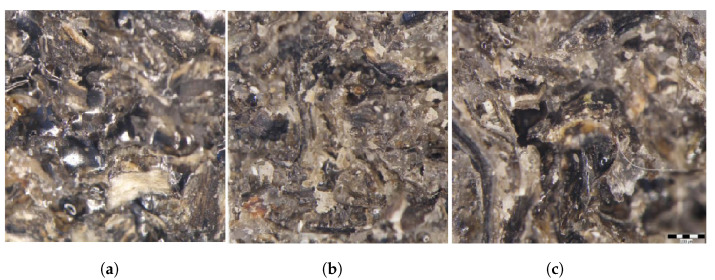
Images with the breaking surfaces of some representative test specimens from the sets: CS0 (**a**); CS1 (**b**); and CS2 (**c**).

**Figure 6 polymers-14-00392-f006:**
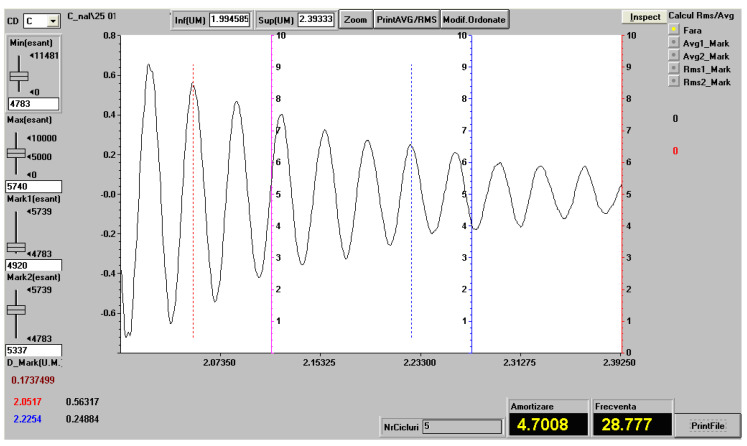
Vibration recording (eigenfrequency and damping factor) in a representative test specimen from the CS2-x set, for the free length of 220 mm.

**Figure 7 polymers-14-00392-f007:**
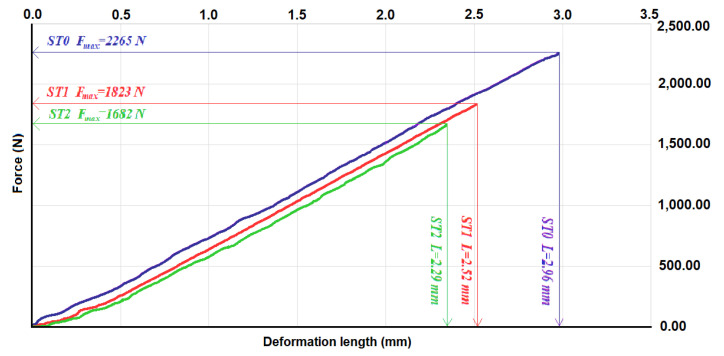
Strength–strain diagram for representative test specimens from sets ST0, ST1 and ST2.

**Table 1 polymers-14-00392-t001:** Characteristics of the sets of test specimens made.

Symbol Set of Test Specimens	Mass Proportion of Resin (%)	Density (g/cm${}^{3})$	Dimensions of Test Specimen (mm)
CS0-x	40	1.15	250 × 25 × 6.7
CS1-x	40	1.11	250 × 25 × 6.8
CS2-x	42	1.07	250 × 25 × 6.8
B0-x	41	1.16	21 × 21 × 21
B1-x	42	1.12	21.1 × 21 × 21
B2-x	44	1.08	21.1 × 21 × 21
ST0-x	41	1.16	250 × 50 × 15
ST1-x	42	1.13	250 × 50 × 15.1
ST2-x	43	1.08	250 × 50 × 15.1

**Table 2 polymers-14-00392-t002:** Average value and standard deviations for modulus of elasticity, tensile strength and elongation at break for CS0-x, CS1-x and CS2-x test specimens.

Test Specimen Type	Modulus of Elasticity *E* (N/mm${}^{2}$)			Tensile Strength $R_{m}$ (MPa)			Elongation at Break *A* (%)		
	Average Value	Linear Average Deviation	Average Square Deviation	Average Value	Linear Average Deviation	Average Square Deviation	Average Value	Linear Average Deviation	Average Square Deviation
	$\bar{x}$	${\bar{d}}_{x}$	$\sigma_{x}$	$\bar{x}$	${\bar{d}}_{x}$	$\sigma_{x}$	$\bar{x}$	${\bar{d}}_{x}$	$\sigma_{x}$
CS0-x	3024	53.9	64.1	9.5	0.28	0.33	0.49	0.022	0.028
CS1-x	2580	51.2	59.8	7.1	0.22	0.24	0.42	0.020	0.022
CS2-x	1814	40	45.6	6.3	0.16	0.18	0.45	0.019	0.021

**Table 3 polymers-14-00392-t003:** Average value and standard deviations for maximum load force and maximum deformation for B0-x, B1-x and B2-x test specimens.

Test Specimen Type	Maximum Load $F_{\max}$ (N)			Maximum Extension at $F_{\max}$ $L_{\max}$ (mm)			Compressive Strength $\sigma_{\mathrm{ac}}$ (MPa)		
	Average Value	Linear Average Deviation	Average Square Deviation	Average Value	Linear Average Deviation	Average Square Deviation	Average Value	Linear Average Deviation	Average Square Deviation
	$\bar{x}$	${\bar{d}}_{x}$	$\sigma_{x}$	$\bar{x}$	${\bar{d}}_{x}$	$\sigma_{x}$	$\bar{x}$	${\bar{d}}_{x}$	$\sigma_{x}$
B0-x	30050	168	210	2.1	0.12	0.14	68.1	1.84	2.16
B1-x	10050	138	164	2.4	0.13	0.14	22.8	0.72	1.11
B2-x	5150	90	103	3.1	0.15	0.18	11.7	0.44	0.52

**Table 4 polymers-14-00392-t004:** Evolution of water absorption and loss, respectively, for the 3 weighed test specimens.

Test Specimen Type	Water Absorption							Water Loss	
	Day 1	Day 2	Day 3	Day 4	Day 5	Day 6	Day 7	Day 8	Day 9
CS0	12.598	12.874	13.155	13.448	13.823	13.990	14.030	13.239	12.611
CS1	13.093	13.745	14.545	15.372	16.210	16.267	16.269	14.512	13.205
CS2	12.643	13.255	13.819	14.475	14.980	15.207	15.257	13.322	12.702

**Table 5 polymers-14-00392-t005:** Vibration behavior of representative test specimens from sets CS0-x, CS1-x and CS2-x.

Free Length (mm)	CS0-x		CS1-x		CS2-x	
	Frequency ν (Hz)	Damping μ (s${}^{-1})$	Frequency ν (Hz)	Damping μ (s${}^{-1})$	Frequency ν (Hz)	Damping μ (s${}^{-1})$
140	89.3	13.1	83.6	15.6	72.2	12.6
160	67.8	9.8	63.4	12.9	55.8	9.8
180	54.4	7.4	49.8	9.5	44.6	7.5
200	44.9	5.4	39.9	7.9	35.4	6.3
220	37.1	4.2	32.8	5.9	28.7	4.7

**Table 6 polymers-14-00392-t006:** Average value and standard deviations for maximum load force and maximum deformation for ST0-x, ST1-x and ST2-x test specimens.

Test Specimen Type	Maximum Load $F_{\max}$ (N)			Maximum Extension at $F_{\max}$ $L_{\max}$ (mm)		
	Average Value	Linear Average Deviation	Average Square Deviation	Average Value	Linear Average Deviation	Average Square Deviation
	$\bar{x}$	${\bar{d}}_{x}$	$\sigma_{x}$	$\bar{x}$	${\bar{d}}_{x}$	$\sigma_{x}$
ST0-x	2255	66	77	2.98	0.11	0.13
ST1-x	1810	50	59	2.47	0.11	0.13
ST2-x	1680	44	52	2.26	0.09	0.11

## Data Availability

The data presented in this study are available on request from the corresponding author.

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
