# Peer review of "A Study of the Mechanical Properties in Composite Materials with a Dammar Based Hybrid Matrix and Reinforcement from Crushed Shells of Sunflower Seeds"

_polymers, 2022, doi:10.3390/polym14030392_

Round 1

Reviewer 1 Report

The reviewer thanks the authors for the submission and the Editor for the invitation. The reviewer feels that the paper is interesting, and it is within the scope of the Journal.

Specific comments:

  • In abstract: if it is true that it is imperative to create new materials with industrial wastes, it is also important that the new materials can be viewed as an alternative to the traditional ones. Hence, the abstract can also present several applications to be used as one alternative. But more important is to give the material and methods, the experimental techniques used and finalize with the best conclusions of the work.
  • In section 1, in line 10, it seems that references 4, 5 and 6 can be merged. In line 45, improve the way the reference 24 is introduced; for instance, authors can say were investigated by Kaymakci et al. The same idea should also be applied to the other references, namely 25, 14, 26, etc…

Please improve the statement “The disadvantage of natural resins is that they form varnishes that require that they be combined with synthetic resins in order to harden.”

The main purpose of this study needs to be described at the end of the introduction. Moreover, because other researchers have also been performing these kinds of studies, authors should emphasis the reasons to perform one more work.

  • In section 2, it seems that line 115 is not a valid paragraph and needs to be joined to the previous paragraph. It seems that resin 1 is related to 60% of natural Dammar while resin 2 is related to 80% of natural Dammar. Nevertheless, the statement presented at lines 110-118 is not clear, and it seems that they have gone a use three types of resin. So, please clarify all these issues at the beginning of section 2.1 and, just after, talk about the differences among them.

Especially this section needs a conscientious English revision. Please, give more information about the procedure used to evaluate the water absorption, how many days the specimens were kept in water, and the water composition. Figure 1 shows the specimens that are described in lines 121-133, but no reference is made to this Figure.

It is not a good idea to present the information of lines 134-141 only in this section. This information is important to the reader and should be presented in the introduction. Moreover, Figure 2 shows the specimen used to create the sandwich composite. Hence some references should be given.

Because the number of different specimens is relatively high, it will be good to present a table with the important information of the several specimens produced in the work.

  • In section 3, The first paragraph of this section is not appropriate; please, improve it. One paragraph with only lines 188 and 189 is not appropriate; please improve the format and arrangement of the paper.

The information presented at lines 242-246 should be given in section 2. In line 264-265, authors mention bar vibration studies and the FSDT, the bar designation is not correct, should be beam vibration. See for instance reference (Augusta Neto, Yu and Roy, 2009) to very the difference between bar and beam.

Augusta Neto, M., Yu, W. and Roy, S. (2009) ‘Two finite elements for general composite beams with piezoelectric actuators and sensors’, Finite Elements in Analysis and Design, 45(5). doi: 10.1016/j.finel.2008.10.010.

The information presented at lines 263-279 is more appropriate in the introduction or material and methods sections. More information should be given to the experimental technique used to measure natural frequencies and damping factor.

  • In section 4, the conclusion in lines 340-342 is based on what? “Are good enough” is not a scientific statement. Please improve.

Author Response

Dear Reviewer 1,

Based on your comments, we have made the following changes to the structure of the paper with

Title: “A STUDY OF THE MECHANICAL PROPERTIES IN COMPOSITE MATERIALS WITH A DAMMAR-BASED HYBRID MATRIX AND REINFORCEMENT FROM CRUSHED SHELLS OF SUNFLOWER SEEDS

Authors: Marius Marinel Stănescu, Alexandru Bolcu*

Responses to specific comments.

  1. In abstract: if it is true that it is imperative to create new materials with industrial wastes, it is also important that the new materials can be viewed as an alternative to the traditional ones. Hence, the abstract can also present several applications to be used as one alternative. But more important is to give the material and methods, the experimental techniques used and finalize with the best conclusions of the work.

Response 1 We have completely modified the abstract as instructed by the reviewers.

  1. In section 1, in line 10, it seems that references 4, 5 and 6 can be merged. In line 45, improve the way the reference 24 is introduced; for instance, authors can say were investigated by Kaymakci et al. The same idea should also be applied to the other references, namely 25, 14, 26, etc…

Response 2 We grouped the references and where appropriate changed the wording (were investigated by ... et al.)

  1. Please improve the statement “The disadvantage of natural resins is that they form varnishes that require that they be combined with synthetic resins in order to harden.”

Response 3 We changed the form of the sentence “Kanehashi and Ishimura showed that natural resins diluted with solvents form varnishes that need to be combined with synthetic resins for hardening.

  1. The main purpose of this study needs to be described at the end of the introduction. Moreover, because other researchers have also been performing these kinds of studies, authors should emphasis the reasons to perform one more work.

Response 4 We have added at the end of the introduction a paragraph in which we have presented the objectives of this study.

  1. In section 2, it seems that line 115 is not a valid paragraph and needs to be joined to the previous paragraph. It seems that resin 1 is related to 60% of natural Dammar while resin 2 is related to 80% of natural Dammar. Nevertheless, the statement presented at lines 110-118 is not clear, and it seems that they have gone a use three types of resin. So, please clarify all these issues at the beginning of section 2.1 and, just after, talk about the differences among them.

Response 5 We have reorganized this section at the request of the reviewers. The line that was 115 we coupled with the previous paragraph. We clarified the three resins (epoxy marked with 0, Dammar 60% marked with 1 and Dammar 80% marked with 2).

  1. Especially this section needs a conscientious English revision. Please, give more information about the procedure used to evaluate the water absorption, how many days the specimens were kept in water, and the water composition. Figure 1 shows the specimens that are described in lines 121-133, but no reference is made to this Figure.

Response 6 We revised English. We have provided some clarification regarding water absorption / loss. In the lines that were 121-133 we referred to figure 1.

  1. It is not a good idea to present the information of lines 134-141 only in this section. This information is important to the reader and should be presented in the introduction. Moreover, Figure 2 shows the specimen used to create the sandwich composite. Hence some references should be given. Because the number of different specimens is relatively high, it will be good to present a table with the important information of the several specimens produced in the work.

Response 7 We also presented information about this section in the introduction. We referred in the text to figure 2. We would introduce a table with the main characteristics of the sets of test pieces made.

  1. In section 3, The first paragraph of this section is not appropriate; please, improve it. One paragraph with only lines 188 and 189 is not appropriate; please improve the format and arrangement of the paper.

Response 8 We have modified the opening paragraph of this section. The former lines 188 and 189 have been added to the next paragraph. We changed the format and reorganized the article.

  1. The information presented at lines 242-246 should be given in section 2. In line 264-265, authors mention bar vibration studies and the FSDT, the bar designation is not correct, should be beam vibration. See for instance reference (Augusta Neto, Yu and Roy, 2009) to very the difference between bar and beam.

“Augusta Neto, M., Yu, W. and Roy, S. Two finite elements for general composite beams with piezoelectric actuators and sensors, Finite Elements in Analysis and Design, 2009, 45(5), 295-304. https://doi.org/10.1016/j.finel.2008.10.010”

Response 9 We gave information from the former lines 242-246 in section 2. In the study of vibration behavior, we used the term beam. We referred to that article.

  1. The information presented at lines 263-279 is more appropriate in the introduction or material and methods sections. More information should be given to the experimental technique used to measure natural frequencies and damping factor.

Response 10 We moved the information presented in the previous lines 263-279 in the introduction. We gave some clarifications for calculating the frequency and damping factor.

  1. In section 4, the conclusion in lines 340-342 is based on what? “Are good enough” is not a scientific statement. Please improve.

Response 11 We modified the former lines 340-342, highlighting the fact that the mechanical properties obtained allow the use of these composites in different fields.

We mention that the answers addressed to reviewer 1 are coloured in the text of the paper in blue, and the answers addressed to all reviewers are coloured in red (the corrected paper is in attachment).                                                

Thanks for the views expressed on the basis of which we have made the changes that have contributed to increasing the scientific level of the paper.

                                                                                      Authors

Reviewer 2 Report

Interesting topic

The mothology is fine

The study is relevant

Author Response

Dear Reviewer 2,

The authors address sincerely thanks the reviewer for the words of appreciation about the scientific quality of the paper:

Title: “ A STUDY OF THE MECHANICAL PROPERTIES IN COMPOSITE MATERIALS WITH A DAMMAR-BASED HYBRID MATRIX AND REINFORCEMENT FROM CRUSHED SHELLS OF SUNFLOWER SEEDS”

Authors: Marius Marinel Stănescu, Alexandru Bolcu*

                                                                                      Authors

Reviewer 3 Report

In this paper, the authors reported composite materials with a dammar based hybrid matrix and reinforcement from crushed shells of sunflower seeds. The paper fit the aims and scope of Polymers. I would recommend accepting the paper after modifications.

  1. Abstract should be updated to reflect the entire research.
  2. Introduction should be substantially improved to clarify the novelty of the manuscript. The author should explain why it is interesting to do the experiments they describe and especially what is new compared to these published papers. The reuse of waste or by-products in agro-food industry can increase economic value and environmental benefits and better highlight sustainability. It might be better to simplify and better explain with realistic examples to evidence the need to reuse agro-food waste by-product.

doi: 10.1016/j.lwt.2021.111617, doi:10.3390/foods9040449, doi: 10.1016/j.ijbiomac.2018.02.018, doi: doi:10.3390/foods8080286, doi:10.3390/polym13132044

  1. Paragraphs 4, 5, 6 and 7 should be reduced.
  2. It was strongly suggested to indicate at the end of the Introduction section the main employed characterisation techniques in order to achieve their purpose.
  3. Section 2 should be reorganized.
  4. Statistical analysis should be performed and described
  5. Significant differences should be mentioned in the Tables.

Author Response

Dear Reviewer 3,

Based on your comments, we have made the following changes to the structure of the paper with

Title: “A STUDY OF THE MECHANICAL PROPERTIES IN COMPOSITE MATERIALS WITH A DAMMAR-BASED HYBRID MATRIX AND REINFORCEMENT FROM CRUSHED SHELLS OF SUNFLOWER SEEDS”

Authors: Marius Marinel Stănescu, Alexandru Bolcu*

  1. Abstract should be updated to reflect the entire research.

Response 1 We have completely modified the abstract as instructed by the reviewers.

  1. Introduction should be substantially improved to clarify the novelty of the manuscript. The author should explain why it is interesting to do the experiments they describe and especially what is new compared to these published papers. The reuse of waste or by-products in agro-food industry can increase economic value and environmental benefits and better highlight sustainability. It might be better to simplify and better explain with realistic examples to evidence the need to reuse agro-food waste by-product.

doi: 10.1016/j.lwt.2021.111617, doi:10.3390/foods9040449, doi: 10.1016/j.ijbiomac.2018.02.018, doi: doi:10.3390/foods8080286, doi:10.3390/polym13132044

Response 2 We have modified the introduction at the request of the reviewers. We have updated the bibliography with the indicated references and we have developed the references to the reuse of waste and by-products from the agri-food industry.

  1. Paragraphs 4, 5, 6 and 7 should be reduced.

Response 3 We rearranged paragraphs 4, 5, 6 and 7. We summarized the contents.

  1. It was strongly suggested to indicate at the end of the Introduction section the main employed characterization techniques in order to achieve their purpose.

Response 4 We have added at the end of the introduction a paragraph in which We have presented the objectives of this study. On the papers we highlighted the techniques and methods used for the study of mechanical properties.

  1. Section 2 should be reorganized.

Response 5 We have reorganized this section at the request of the reviewers.

  1. Statistical analysis should be performed and described. Significant differences should be mentioned in the Tables.

Response 6 We gave the calculation formulas for the values ​​of linear mean deviation and quadratic mean deviation and entered in the tables the data of the experimental determinations from the tensile, compressive and bending stresses. The maximum number of experimental determinations for a set of test pieces was 15, which is a limited value for a statistical study.

We mention that the answers addressed to reviewer 3 are colored in the text of the paper in green, and the answers addressed to all reviewers are colored in red (the corrected paper is in attachment).

Thanks for the views expressed on the basis of which we have made the changes that have contributed to increasing the scientific level of the paper.                                                                                                         Authors

Round 2

Reviewer 1 Report

thanks to the Authors for the paper improvement. 

Reviewer 3 Report

All the comments had been addressed.